# Bone Health in Patients with Rheumatoid Arthritis in Bahrain

**DOI:** 10.3390/medicina60122078

**Published:** 2024-12-18

**Authors:** Adla B. Hassan, Amer Almarabheh, Abdulaziz Almekhyal, Danya Abdulhameed AlAwadhi, Haitham Jahrami

**Affiliations:** 1College of Medicine and Health Sciences, Arabian Gulf University, Manama 329, Bahrain; amerjka@agu.edu.bh (A.A.); almekhyal132@hotmail.com (A.A.); hjahrami@health.gov.bh (H.J.); 2King Abdullah Medical City, Manama 329, Bahrain; 3Government Hospitals, Manama 329, Bahrain; dania.alawadhi@hotmail.com

**Keywords:** bone mass density, RA, vitamin D, DMT2, uric acid, Ca breast

## Abstract

*Background and Objectives*: Compared to the general population, rheumatoid arthritis (RA) patients have additional disease-specific risk factors for osteoporosis that include chronic exposure to systemic inflammation. The current study aimed to investigate the prevalence of osteoporosis and its associated risk factors, such as age, sex, body mass index (BMI), uric acid (UA), and vitamin D status, but also the coexistence of type 2 diabetes mellitus (DMT2) and breast cancer (Ca breast) in patients with RA in Bahrain. *Material and Methods*: Data from DEXA scans were collected retrospectively from the patient’s electronic health records. All patients who had BMD data and at least one single comorbidity, including RA, were included in the current study. The collected data were analyzed by using SPSS, version 28. *Results*: A total of 4396 patients were included in the current study. The comorbidities among this cohort were as follows: 3434 patients had endocrinological diseases, among them 63.6% had DMT2; 1870 patients had rheumatological diseases, among them 15.1% had rheumatoid arthritis; and 941 patients had malignancies, among them 75.6% had breast cancer. Our results indicated that patients with RA had a high prevalence of low BMD (72.30%, *p* < 0.001) and low vitamin D levels (63.10%, *p* < 0.001) but high serum UA (20.85%). Comparing RA with non-RA patients, our results showed a statistically significant association between RA and each of BMD and UA (*p* = 0.017 and *p* = 0.004, respectively), but also between RA and each of age (*p* = 0.001) and Ca breast (*p* < 0.001). However, no association was found between RA and BMI, DMT2, or vitamin D status. *Conclusions*: RA patients had a high prevalence of low BMD (72.3%) and low vitamin D (63.10%) but high serum UA (20.85%). The risk of osteoporosis, hypovitaminosis, and gout must be kept in mind during the evaluation of any case with RA.

## 1. Introduction

Rheumatoid arthritis (RA) is commonly associated with decreased bone mineral density (BMD) or osteoporosis, which is a well-established extra-articular feature of this disease [1,2,3]. However, the mechanism behind bone loss is not well understood. A recent study showed that RA patients exhibit deficits in cortical bone structure and trabecular density at the tibia and a preserved functional muscle-bone unit [4]. Moreover, a possible role of vitamin D and its receptor gene polymorphisms in the etiopathogenesis of RA-associated osteoporosis has been reported [5]. Similar to the general population, osteoporosis in RA could be due to numerous factors that might be considered as risk factors for bone loss, such as body mass index (BMI), female > 50 years, smoking, glucocorticoid exposure, and 25-hydroxy vitamin D (vitamin D) deficiency [6]. Vitamin D deficiency is a major risk factor for accelerated bone loss and fracture [7]. Vitamin D deficiency has a high risk for the development of RA [8]; similarly, increased vitamin D intake has a lower risk of developing RA [9].

In addition to the abovementioned risk factors that affect the general population, RA patients have disease-specific risk factors for osteoporosis and fractures, which include pathogenic anticitrullinated peptide antibody, chronic exposure to systemic inflammation, and joint damage causing early disability [10,11]. The early phase of RA is very critical because it is the inflammatory phase. Therefore, the rate of erosion, number of inflamed and tender joints, and rate of bone loss are also at their peak during this time. If the patient is left untreated during the early phase of RA, deterioration and hand deformity may occur. Initiation of therapy at this phase is of great benefit to reduce the activity and improve the radiographic outcomes in RA patients [12,13]. In RA patients, a population-based study indicated that adequate management of patients with RA, addressing both the anti-rheumatic disease and anti-osteoporosis therapy, protects against bone loss [1]. However, BMD, which is determined using a DEXA scan, is not routinely performed in Bahrain and will be requested only for those with chronic diseases, such as type 2 diabetes mellitus (DMT2), vitamin D deficiency, or long-term corticosteroid therapy, or with a history of fracture, but not with those with gout. Typically, in patients with RA, the data on the relation between BMD and fracture rates are limited. However, a recent Canadian study revealed that although compliance with current Canadian osteoporosis guidelines for patients with RA remains low among all rheumatologists, the higher-risk patients, according to the 10-year fracture risk tool, were more likely to have a BMD test and receive treatment for osteoporosis [14].

Regarding serum uric acid (UA), a longitudinal study revealed that patients with elevated serum UA (gout) had a 20% increase in the risk of developing osteoporosis in the future [15]. Furthermore, the coexistence of elevated serum UA and vitamin D deficiency has been reported previously in healthy adults and the elderly [16,17]. However, the coexistence of gout and RA is relatively rare, which could be due to lacking studies in this area. In DMT2 patients in Bahrain, the higher the BMI, the higher the BMD, and the lesser the risk of developing osteoporosis, but RA was not included in that study [18]. Similarly, in Saudi male patients, the presence of DMT2 did not influence or increase the incidence of osteoporosis [19]. The high BMI was strongly associated with increased risk for RA, with weight gain of ≥20 kg associated with more than a three-fold increased risk of RA [20,21]. High BMI, although associated with high BMD, is not protective against fracture in postmenopausal women [22]. Regarding Ca breast, many studies suggested that vitamin D deficiency carries a major risk for developing cancer in women, and daily ingestion of 1100 IU vitamin D3 for 4 years reduced the risk of developing cancer by more than 60% [23]. Methotrexate, which is the drug of choice for the treatment of RA, has been used at much higher doses to treat breast cancer compared to RA. In patients with RA, methotrexate does not increase the risk of developing cancer. However, because cancer immunotherapy activates the immune system, it may worsen the symptoms of RA. Furthermore, breast cancer therapy often leads to estrogen deprivation with a subsequent increased risk of low BMD and fracture [24].

In the Kingdom of Bahrain, neither screening for BMD nor screening for vitamin D or UA is routinely done at health services for patients with RA. Thus, the nationwide prevalence of osteoporosis in Bahraini patients with RA has not been investigated. The current study aimed to conduct a retrospective cross-sectional study to investigate the prevalence of osteoporosis and/or osteopenia in adult Bahraini patients with RA. We also aimed to investigate the risk factors for bone loss, such as age, sex, serum UA, vitamin D3 levels, BMI, and DMT2 in patients with RA in Bahrain.

## 2. Materials and Methods

### 2.1. Patients

The current study is a retrospective cross-sectional study. The demographic, anthropometric, clinical, and pharmacy data from the electronic medical health records was collected. Data from DEXA scans were collected retrospectively from patients’ electronic medical health records who underwent scans for the diagnosis of osteoporosis during a period of three years from the beginning of 2016 to the end of 2018. The current study is an extension of a big project in which a detailed method was described [25,26]. The data were collected from three big hospitals or centers in the Kingdom of Bahrain: The University Medical Center/King Abdallah Medical City (KAMC), Salmanyia Medical Center (SMC), and King Hamed University Hospital (KHUH). Patients’ medical records were reviewed for the DEXA data; demographic, anthropometric, clinical, and pharmacy data; comorbidities; as well as any related laboratory investigations. The patients with RA were diagnosed according to 2010 ACR/EULAR criteria [27] and were followed on a regular basis at the rheumatology clinics at the three mentioned hospitals.

### 2.2. Methods

BMD measurements were done using Dual Energy X-ray A absorptiometry (DEXA) at SMC, KHUH, and KAMC. The diagnosis of osteopenia (T score ≤ −1 and >−2.5) and/or osteoporosis (T score ≤ −2.5) were as WHO criteria [28]. The measurement of the serum level of 25 (OH) cholecalciferol, vitamin D (D3), was estimated using chemiluminescence immunoassay on Advia Centaur Analyzer (LoD 8.0 nmol/L). Serum level between 30 nmol/L and 50 nmol/L (≥30 ˂50) was classified as vitamin D insufficiency, level < 30 nmol/L was considered as vitamin D deficiency, and the optimal level was defined as ≥50 nmol/L. UA was determined in serum by using the uricase (enzymatic) method. The reference range of serum UA was defined as 220–547 for males and 184–464 micromole/L for females. Heights, weights, and calculated BMI were routinely recorded with the patient’s electronic files. Comorbidities such as DMT2 and Ca breast were collected from the patient’s electronic files if recorded.

## 3. Statistical Analysis

Categorical variables were represented as frequencies and percentages, whereas continuous variables were represented as mean and standard deviation. The chi-square test was utilized to show any potential association between the status of RA with BMD overall and with other patient characteristics: age, sex, BMI, vitamin D, UA, DMT2, and Ca breast. Additionally, we computed the odds ratio (OR) to further evaluate the strength of the associations observed. The collected data were analyzed by using the Statistical Package for Social Sciences (SPSS), version 28. A *p*-value of less than 0.05 was considered statistically significant.

### Ethical Considerations

This research was initiated after obtaining ethical approvals from all research and ethical committees at the Arabian Gulf University (AGU) for King Abdullah Medical city (KAMC), Secondary Health Care Research Sub Committee at Salmaniya Medical Complex (SMC), and King Hamad University Hospital (KHUH) (No: E003-PI-10/18, & No 273/2019).

## 4. Results

### 4.1. Sample Characteristics

A total of 6,307 patients’ health records were reviewed retrospectively from the three hospitals; more than three-quarters of our data was extracted from SMC 79.5% (5016), while the rest were from KHUH 15.7% (991) and KAMC 4.8% (300). The comorbidities among the studied cohort were reported as follows: endocrinological diseases were 71% (3434 patients). Among them, DMT2 was 63.6% (2183), followed by rheumatological diseases at 15.1% (1870 patients), and among them, RA was 14.7% (282). Malignancy was 19.4% (941 patients), and among them, Ca breast was 75.6% (712). Unfortunately, some patients share two or three diseases together. Due to missing data, only 4,396 patients had complete BMD data, and at least one single comorbidity of either RA, DMT2, or Ca breast was included for final analysis in the current study.

### 4.2. Demographic Characteristics and Data Associated with Rheumatoid Arthritis Patients Compared to Non-Rheumatoid Arthritis Patients

The results of the chi-square test in Table 1 showed that when comparing RA patients with non-RA patients, there was a statistically significant association between RA and each of BMD (at any site femur or lumbar spine) (χ^2^ = 5.722, df = 1, *p* = 0.017), UA (χ^2^ = 8.305, df = 1, *p* = 0.004), age (χ^2^ = 18.335, df = 1, *p* = 0.001), and Ca breast (χ^2^ = 13.144, df = 1, *p* < 0.001). The age group analysis revealed a statistically significant association with RA prevalence (*p* = 0.001), with notable variations in RA risk across different age categories. Younger individuals (<44 years) demonstrated the lowest RA prevalence, while middle-aged and older groups showed progressively increased RA risk. BMI showed a significant association with RA risk (OR = 1.42, *p* = 0.017), with patients having abnormal (low) BMD showing a 42% higher likelihood of developing RA. UA levels demonstrated a statistically significant relationship with RA occurrence (OR = 1.58, *p* = 0.004), with individuals having normal UA levels showing 58% higher odds of developing RA. Ca breast status exhibited a highly significant association with RA risk (OR = 0.36, *p* < 0.001), with RA patients showing substantially lower odds (64% reduction) of Ca breast compared to the non-RA group.

Similar results are shown in Table 2; when we compared RA patients to all other rheumatic and musculoskeletal diseases (RMD), we found a statistically significant association between RA and all parameters except for age and BMD. Vitamin D status showed a significant association with RA (OR = 1.54, *p* = 0.009), with patients having non-optimal vitamin D levels demonstrating 54% higher odds of developing RA compared to those with optimal levels. The BMI analysis revealed a statistically significant relationship with RA risk (*p* < 0.001), with underweight individuals showing the highest odds of RA (OR = 2.79), normal-weight individuals at OR = 2.33, and overweight patients at OR = 1.40. DMT2 demonstrated a statistically significant association with RA prevalence (OR = 0.76, *p* = 0.041), with DMT2 patients showing 24% lower odds of developing RA. UA levels exhibited a significant relationship with RA occurrence (OR = 1.67, *p* = 0.003), with normal UA levels associated with 67% higher odds of RA. Ca breast status showed a statistically significant association with RA risk (OR = 0.54, *p* = 0.037), with Ca breast patients demonstrating 46% lower odds of developing RA.

### 4.3. Data Associated with Bone Mineral Density (BMD)

Due to missing data, only 4396 patients had complete BMD data. Our results in Table 3 indicated that there were statistically significant associations between the status of BMD and each of RA, sex, postmenopausal, DMT2, and UA (χ^2^ = 5.772, df = 1, *p* = 0.017; χ^2^ = 12.252, df = 1, *p* < 0.001; χ^2^ = 32.525, df = 2, *p* < 0.001; χ^2^ = 4.277, df = 1, *p* = 0.039; χ^2^ = 17.876, df = 1, *p* < 0.001), respectively.

### 4.4. Data Associated with Rheumatoid Arthritis Patients

Our results, depicted in Table 4, Figure 1 and Figure 2, indicated that the patients with RA had a high prevalence of low BMD (72.30%), which was statistically significant (*p* < 0.001). Similarly, RA patients had a high prevalence of low vitamin D levels (63.10%), which was statistically significant (*p* < 0.001). The prevalence of high serum UA was 20.85% (Table 4). Our results also revealed a high prevalence of female gender, 95.38% (*p* < 0.001), and age of 60 years old and above 60.1% (*p* < 0.001) among RA patients. Our results also showed that within the 282 RA cohort, 46 patients were seropositive (16.4%), and 25 were seronegative (8.9%), while 210 (74.7%) just reported as RA without defining them as seropositive or seronegative.

### 4.5. Association of Serum Uric Acid with RA, DMT2, and Ca Breast

Our results revealed that UA was associated with both RA (Table 1) and DMT2 (*p* = 0.004 and *p* < 0.001, respectively) but not with Ca breast (*p* = 0.139).

### 4.6. Data Associated with Vitamin D

Regarding the level of vitamin D, the results (Appendix A) revealed there was a statistically significant association between the status of vitamin D and each of postmenopausal and Ca breast (χ^2^ = 8.912, df = 2, *p* =0.012; χ^2^ = 4.687, df = 1, *p* = 0.030, respectively).

### 4.7. Data Associated with Body Mass Index (BMI)

Regarding the body mass index (BMI), the results (Appendix A) revealed there was a statistically significant relationship between BMI and all factors (DM, Ca breast, BMD, and UA). The probability values (*p* < 0.001) for all these factors were statistically significant.

## 5. Discussion

RA patients have additional disease-specific risk factors for osteoporosis or bone loss compared to normal populations [10]. In the current study, the aim was to conduct a retrospective cross-sectional study to investigate the prevalence of low BMD in adult Bahraini patients with RA and its association with comorbidities such as DMT2 and Ca breast. Our objective was also to investigate the risk factors for bone loss or reduced BMD, such as age, sex, BMI (obesity), UA, and vitamin D status in patients with RA. The results showed that RA patients had a high prevalence of low BMD (72.3%) and low vitamin D (63%) but high serum UA (20.85%). Furthermore, when comparing RA patients with non-RA patients, the results showed a statistically significant association between RA and each of BMD and UA.

The current study is an extension and last manuscript of a big project with already published articles [25,26]. Generally, in Bahrain, all basic investigations would be done routinely as part of health services for patients with chronic diseases. However, BMD, which is determined using a DEXA scan, is not routinely performed and will be requested only for patients with certain chronic diseases such as vitamin D deficiency, a history of fracture, or long-term corticosteroid therapy. Therefore, in any retrospective hospital-based study investigating BMD, one cannot exclude the comorbidities. Preliminary analysis of the whole BMD cohort showed a statistically significant association between BMD and each of the patient’s characteristics: RA, sex, UA, and DMT2, but not with Ca breast or vitamin D status. The lack of association between BMD and Ca breast could be explained by the fact that the Ca breast cohort was already on anti-osteoporotic therapy, including vitamin D therapy. Among all rheumatological diseases in this cohort, RA disease was found to be 15.1%; thus, 282 RA patients were included in the current study. Patients with RA are at increased risk for osteoporosis, cancer, serious infection, lung disease, and cardiovascular disease than the general population [29]. The current study showed that when comparing RA patients with non-RA patients, there was a statistically significant association between RA and each BMD, UA, age, and Ca breast. Similar results were obtained in Table 2, when RA was compared with other RMD (1870 patients), we found statistically significant associations between RA and all parameters investigated in this study, except for age and BMD. This finding could be explained by the fact that most of the RMD occur in young and middle-aged females and their RMD put them at increased risk of low BMD.

Further analysis within the RA cohort revealed that RA patients had a high prevalence of low BMD (72.3%) and low vitamin D (63%) but high serum UA (20.85%). To our knowledge, this is the first study investigating the prevalence of low BMD in Bahraini patients with RA. Our result of low BMD in RA is consistent with a recent study from three northwestern European countries that demonstrated low BMD in RA but also an association between low BMD and radiographic RA damage [30]. The coexistence of gout and RA is relatively rare, which could be due to a lack of studies in this area. However, in Bahrain, RA and gout have not been reported before, except in one previous study completed by our team, which showed that RA patients had hypovitaminosis and high serum UA levels. The correction of hypovitaminosis with vitamin D3 therapy led to a reduction of serum UA in Bahraini RA patients; nevertheless, that was a small cohort of only 30 RA patients, and BMD data was not included in that study [31].

Unfortunately, the present study revealed that among all malignancies reported in this studied cohort, the Ca breast accounted for 70.4%. Furthermore, the prevalence of Ca breast among RA patients was 4.61%. It is known that patients with Ca breast had significant bone loss and subsequent fracture [24]. Our result of the association of RA with Ca breast is consistent with a previous study that showed RA disease and its therapy was a risk factor for cancer development [29]. Furthermore, our results revealed that the prevalence of obesity among RA patients was very high (56.09%), which was statistically significant, while the prevalence of DMT2 among RA patients was 37.94%. The association of our RA cohort with high BMI and low BMD could only be explained if there were so many factors contributing to bone loss in RA patients.

We recommended routine measurement of BMD and all parameters involved in the regulation of bone health, including serum vitamin D and serum UA, to be requested routinely as part of RA health services. Due to seasonal variations, vitamin D serum levels should be measured at least twice yearly for any patient with RA [32]. Usually, measurement of BMD is only useful if it can effectively be followed by osteoporotic therapy. A German survey has shown deficits in the administration of osteoporosis medication, especially in premenopausal women, although 50% of them were diagnosed as having osteoporosis [33]. Similarly, in the Kingdom of Bahrain, patients with osteoporosis were undertreated with anti-resorptive drugs [25].

To enhance the robustness of our findings, future studies should prioritize conducting meta-analyses of individual participant data, alongside exploring other innovative research methodologies. These approaches will facilitate the replication of results across diverse cohorts, thereby increasing the generalizability of our findings. Moreover, by integrating various research designs, we can deepen our understanding of the identified factors, ultimately contributing to a more comprehensive and nuanced perspective in the field. Such efforts will not only validate our current results but also pave the way for future research initiatives.

### Limitations and Strengths of the Study

Our study had certain limitations. As a retrospective, there was some missing data in the electronic records. Thus, more than 2000 subjects were excluded from the first step in the analysis. More patients were excluded, as more selection of the cohort based on a patient should have at least one single comorbidity. Additionally, the medications used by those patients, as reported in their electronic files, could also affect the results of our risk factors, such as UA and vitamin D status. Other potential limitations were missing data on other risk factors, such as smoking, diet, and lifestyle. One more limitation is that to see the actual prevalence, we need to have country-specific reference data, which is completely lacking. On the other hand, the strength of our study is that we included a large sample of 282 RA patients relative to a small country such as Bahrain. In the absence of nationwide studies investigating chronic diseases such as RA, this multicenter study provides valuable data about the prevalence of osteoporosis in RA disease in Bahrain.

## 6. Conclusions

RA patients had a high prevalence of low BMD (72.3%) and low vitamin D (63.10%) but high serum UA (20.85%). Vitamin D deficiency is a risk factor for low BMD but also a risk factor for high serum UA. Similarly, high serum UA is a risk factor for low BMD. Unfortunately, the coexistence and the interplay of these three factors in RA patients will make the condition worse if undetected early. Moreover, recognition of acute gout in patients with RA may be difficult or mistaken for arthritis of RA. Therefore, the risk of osteoporosis, hypovitaminosis, and gout must be kept in mind during the evaluation of any case with RA. We recommended follow-up with serial measurements of serum UA, serum vitamin D levels, and BMD routinely as part of RA health services for confirmation of the diagnosis of gout, hypovitaminosis, and osteoporosis, thus early intervention to prevent RA-associated radiographic damage.

## Figures and Tables

**Figure 1 medicina-60-02078-f001:**
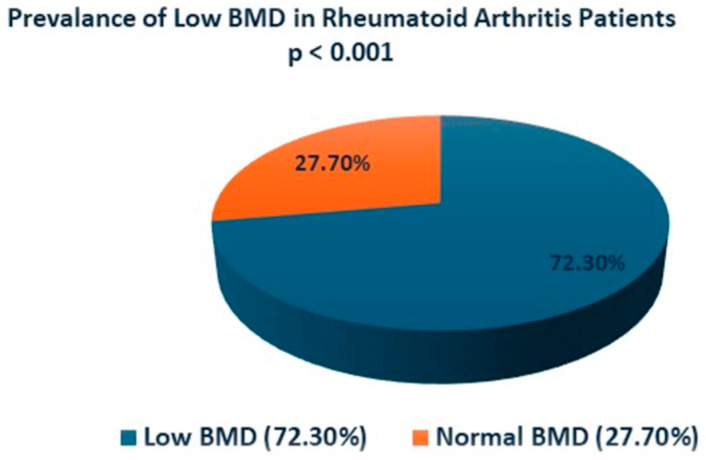
Prevalence of low BMD in RA patients.

**Figure 2 medicina-60-02078-f002:**
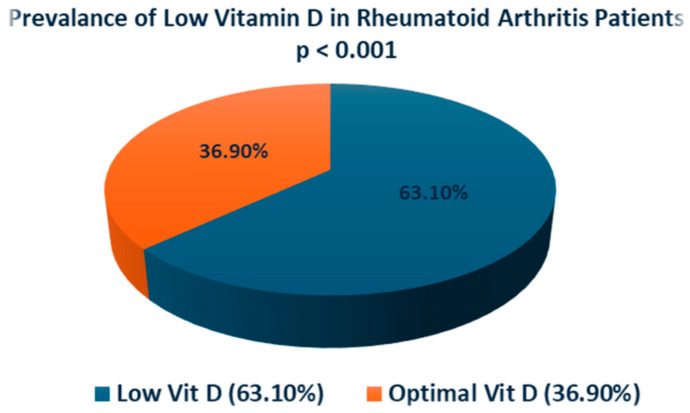
Prevalence of low vitamin D in RA patients.

**Table 1 medicina-60-02078-t001:** Demographic characteristics and data associated with RA patients compared to non-RA patients.

Characteristic	Patients		chi-Square	*p*-Value
	Non-RA n (%)	RA n (%)		
**Gender**				
Male	442 (97.1)	13 (2.9)	3.547	0.060
Female	5328 (95.2)	268 (4.8)		
**Age Group**				
<44 Years	479 (98)	10 (2)	18.335	0.001
44–<56 Years	1364 (96.5)	49 (3.5)		
56–<60 Years	962 (94.8)	53 (5.2)		
60–<75 Years	2454 (94.4)	146 (5.6)		
=>75 Years	513 (95.7	23 (4.3)		
**BMD**				
Normal	1469 (95.8)	65 (4.2)	5.722	0.017
Abnormal (low BMD)	2692 (94.1)	170 (5.9)		
**Vitamin D**				
Non-optimal	1653 (92.7)	130 (7.3)	0.0777	0.782
Optimal	927 (92.4)	76 (7.6)		
**BMI**				
Underweight	81 (92.0)	7 (8.0)	6.627	0.085
Normal	700 (93.6)	48 (6.4)		
Overweight	1285 (94.6)	74 (5.4)		
Obesity	2184 (95.6)	101 (4.4)		
**DMT2**				
Yes	2075 (95.1)	107 (4.9)	1.466	0.226
No	3951 (95.8)	175 (4.2)		
**Uric acid**				
Normal	2568 (93.2)	186 (6.8)	8.305	0.004
High	1083 (95.7)	49 (4.3)		
**Ca breast**				
Yes	699 (98.2)	13 (1.8)	13.144	< 0.001
No	5327 (95.2)	269 (4.8)		

BMD: bone mineral density. BMI: body mass index. DMT2: type 2 diabetes mellitus. RA: rheumatoid arthritis. Percentages presented in the tables are calculated per row.

**Table 2 medicina-60-02078-t002:** Demographic characteristics and data associated with RA arthritis patients compared to other RMD.

Characteristic	Patients		chi-Square	*p*-Value
	RMA n (%)	RA n (%)		
**Gender**				
Male	95 (87.2)	14 (12.8)	0.435	0.582
Female	1493 (84.8)	267 (15.2)		
**Age Group**				
<44 Years	76 (89.4)	9 (10.6)	1.898	0.754
44–<56 Years	292 (85.4)	50 (14.6)		
56–<60 Years	279 (83.8)	54 (16.2)		
60–<75 Years	801 (84.7)	145 (15.3)		
=>75 Years	140 (85.9)	23 (14.1)		
**BMD**				
Normal	379 (85)	65 (4.2)	2.98	0.08
Abnormal (low BMD)	754 (81.5)	170 (5.9)		
**Vitamin D**				
Non-optimal	390 (74.7)	132 (25.3)	6.899	0.009
Optimal	343 (81.9)	76 (18.1)		
**BMI**				
Underweight	28 (80.0)	7 (20.0)	16.824	<0.001
Normal	175 (78.5)	48 (21.5)		
Overweight	376 (83.7)	73 (16.3)		
Obesity	784 (88.4)	103 (11.6)		
**DMT2**				
Yes	701 (86.9)	106 (13.1)	4.194	0.041
No	887 (83.4)	176 (16.6)		
**Uric acid**				
Normal	860 (82.1)	187 (17.9)	8.904	0.003
High	375 (88.4)	49 (11.6)		
**Ca breast**				
Yes	130 (90.9)	13 (9.1)	4.337	0.037
No	1458 (84.4)	269 (15.6)		

RMA: rheumatic and musculoskeletal diseases. RA: rheumatoid arthritis. BMI: body mass index. BMD: bone mineral density. DMT2: diabetes mellitus type 2. Ca breast: cancer of the breast. Percentages presented in the tables are calculated per row.

**Table 3 medicina-60-02078-t003:** Data associated with BMD.

Characteristic	BMD Overall		chi-Square	*p*-Value
	Normal n (%)	Low n (%)		
**Gender**				
Male	149 (43.6)	193 (56.4)	12.252	< 0.001
Female	1385 (34.2)	2668 (65.8)		
**Postmenopausal**				
56–<60 Years	253 (33.6)	501 (66.4)	32.525	< 0.001
60–<75 Years	512 (28.8)	1265 (71.2)		
=>75 Years	67 (17.4)	317 (82.6)		
**Vitamin D**				
Non-optimal	568 (35.4)	1038 (64.6)	1.425	0.233
Optimal	267 (32.9)	544 (67.1)		
**DMT2**				
Yes	582 (36.9)	996 (63.1)	4.277	0.039
No	952 (33.8)	1866 (66.2)		
**Uric acid**				
Normal	611 (30.0)	1428 (70.0)	17.876	< 0.001
High	305 (38.2)	493 (61.8)		
**Ca breast**				
Yes	229 (37.4)	384 (62.6)	1.900	0.168
No	1305 (34.5)	2478 (65.5)		
**RA**				
Yes	65 (27.7)	170 (72.3)	5.772	0.017
No	1469 (35.3)	2692 (64.7)		

BMD: bone mineral density. DMT2: type 2 diabetes mellitus. RA: rheumatoid arthritis. Percentages presented in the tables are calculated per row.

**Table 4 medicina-60-02078-t004:** Data associated with RA patients.

Characteristics	RA N (T = 282)	RA %	*p*-Value
**Gender**			
Male	13	4.62	<0.001
Female	268	95.38	
**Age Group**			
<60 Years	112	39.90	<0.001
=>60 Years	169	60.10	
**BMD**			
Normal	65	27.70	<0.001
Abnormal (low BMD)	170	72.30	
**Vitamin D**			
Non-optimal	130	63.10	<0.001
Optimal	76	36.90	
**BMI**			
Obesity	129	56.09	0.009
Non-obesity	101	43.91	
**DMT2**			
Yes	107	37.94	<0.001
No	175	62.06	
**Uric acid**			
Normal	186	79.15	<0.001
High	49	20.85	
**Ca breast**			
Yes	13	4.61	<0.001
No	269	95.39	

BMD: bone mineral density. BMI: body mass index. DMT2: type 2 diabetes mellitus. RA: rheumatoid arthritis. T: total. Percentages presented in the tables are calculated per column.

## Data Availability

Data are available upon request from the corresponding author.

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
