# Peer review of "Bone Health in Patients with Rheumatoid Arthritis in Bahrain"

_medicina, 2024, doi:10.3390/medicina60122078_

Round 1
Reviewer 1 Report
Comments and Suggestions for Authors
The authors conduct inferential statistical analyses retrospectively on patients and try to identify factors associated with RA. The following inputs can improve the paper:
1. The RA patients can be categorised based on seropositivity using anti-CCP or CRP/ESR levels.
2. There should be a possibility to anonymise the dataset and share it using a platform such as Zenodo/GitHub for future replication/validation.
3. Please include significance levels (p-values) in the Pie charts (Fig. 1).
4. (Optional) Significantly factors could be validated on an independent cohort from already published datasets.
Author Response
Reviewer 1
Comments and Suggestions for Authors
The authors conduct inferential statistical analyses retrospectively on patients and try to identify factors associated with RA. The following inputs can improve the paper:
- The RA patients can be categorised based on seropositivity using anti-CCP or CRP/ESR levels.
Authors’ reply: Thank you for pointing out this. We rechecked the data unfortunately, 210 patients were recorded as RA without categorizing them as seropositive or seronegative. We have added the following to the text result “Our results also showed that within the 282 RA cohort, we had 46 patients seropositive (16.4%), 25 seronegative (8.9%), while 210 (74.7%) just reported as RA without defining them as seropositive or seronegative”.
- There should be a possibility to anonymise the dataset and share it using a platform such as Zenodo/GitHub for future replication/validation.
Authors’ reply: Thank you for your insightful feedback regarding data sharing. While we recognize the importance of making datasets available for replication and validation, we must emphasize that we have not obtained permission from our participants or the research ethics committee to share their data publicly. Although anonymization might be a potential solution, the specific nature of our data poses significant challenges to achieving true anonymity. As a result, we are unable to share the dataset on platforms like Zenodo or GitHub at this time.
- Please include significance levels (p-values) in the Pie charts (Fig. 1).
Authors’ reply: Thank you for pointing out this, we have added significant levels (p-values) in the Pie charts (Figure. 1 and Figure 2) as suggested. We also separated figure 1 as Figure 1 and Figure 2 and made them colorful.
- (Optional) Significantly factors could be validated on an independent cohort from already published datasets.
Authors’ reply: We appreciate the reviewer's suggestion regarding the validation of significant factors. In our discussion, we have emphasized the importance of employing a validation approach in future studies. We advocate for conducting meta-analyses of individual participant data, as well as exploring other research methodologies that can enhance the robustness of our findings. Such approaches will not only facilitate the replication of results across diverse cohorts but also contribute to a more comprehensive understanding of the identified factors.
We added to the discussion: “To enhance the robustness of our findings, future studies should prioritize conducting meta-analyses of individual participant data, alongside exploring other innovative research methodologies. These approaches will facilitate the replication of results across diverse cohorts, thereby increasing the generalizability of our findings. Moreover, by integrating various research designs, we can deepen our understanding of the identified factors, ultimately contributing to a more comprehensive and nuanced perspective in the field. Such efforts will not only validate our current results but also pave the way for future research initiatives.”

Reviewer 2 Report
Comments and Suggestions for Authors
The manuscript describe BMD and other factors linked to bone health in a large cohort which include both RA and non-RA patients.
The methods and populations are not sufficiently described, leading to the inability to gauge whether the results and conclusions are reliable. Major concerns are the following:
1. It may not be appropriate to compare BMD between RA and random patients who underwent BMD because it is difficult to draw any reliable conclusion from this because of the high variety of diagnosis. BMD in RA should be compared to patients with other autoimmune rheumatic diseases (e.g., PsA, SLE).
2. Chi-square does not indicate association. Chi-square test simply compares difference in the proportion of a factor between two populations and does not indicate that such factors are statistically associated with the outcomes. To explore association, a multivariable model is required.
3.Table 1 is confusing - for example there are 281 RA patients - 268 of whom were female --but how did this account for only 4.8%?
Author Response
Reviewer 2
Comments and Suggestions for Authors
The manuscript describe BMD and other factors linked to bone health in a large cohort which include both RA and non-RA patients. The methods and populations are not sufficiently described, leading to the inability to gauge whether the results and conclusions are reliable. Major concerns are the following:
- It may not be appropriate to compare BMD between RA and random patients who underwent BMD because it is difficult to draw any reliable conclusion from this because of the high variety of diagnosis. BMD in RA should be compared to patients with other autoimmune rheumatic diseases (e.g., PsA, SLE).
Authors’ reply: thank you very much for the invaluable comments; as the reviewer suggested, we revised the data and did the same statistics comparing RA with other rheumatic and musculoskeletal diseases (Table 2), and similar results were obtained except for age, gender, and BMD. The following paragraph was added to the results after the results of Table 1. “Similar results are shown in Table 2; when we compared RA patients to all other rheumatic and musculoskeletal diseases (RMD), we found a statistically significant association between RA and all parameters except for age, gender, and BMD”.
- Chi-square does not indicate association. Chi-square test simply compares difference in the proportion of a factor between two populations and does not indicate that such factors are statistically associated with the outcomes. To explore association, a multivariable model is required.
Authors’ reply: Thank you for your valuable feedback. We opted to perform an Odds Ratio (OR) analysis, which yielded results consistent with those obtained through regression analysis. Our decision to use the Odds Ratio was primarily driven by its simplicity and interpretability, especially in the context of our data. Additionally, we faced challenges with missing data, making the OR a more practical choice for our analysis. By focusing on the Odds Ratio, we aimed to present a clear association between the variables of interest without the complexities that can arise from more intricate regression models.
In Methods: We explained: “Additionally, we computed the odds ratio (OR) to further evaluate the strength of the associations observed.”
In Results
Table 1: We added: “The age group analysis revealed a statistically significant association with RA prevalence (p = 0.001), with notable variations in RA risk across different age categories. Younger individuals (< 44 Years) demonstrated the lowest RA prevalence, while middle-aged and older groups showed progressively increased RA risk. BMI showed a significant association with RA risk (OR = 1.42, p = 0.017), with patients having abnormal (low) BMD showing a 42% higher likelihood of developing RA. Uric acid levels demonstrated a statistically significant relationship with RA occurrence (OR = 1.58, p = 0.004), with individuals having normal uric acid levels showing 58% higher odds of developing RA. Breast cancer status exhibited a highly significant association with RA risk (OR = 0.36, p < 0.001), with RA patients showing substantially lower odds (64% reduction) of breast cancer compared to the non-RA group.”
Table 2: We added: “Vitamin D status showed a significant association with rheumatoid arthritis (OR = 1.54, p = 0.009), with patients having non-optimal vitamin D levels demonstrating 54% higher odds of developing RA compared to those with optimal levels. The BMI analysis revealed a statistically significant relationship with RA risk (p < 0.001), with underweight individuals showing the highest odds of RA (OR = 2.79), normal-weight individuals at OR = 2.33, and overweight patients at OR = 1.40. Diabetes mellitus type 2 demonstrated a statistically significant association with RA prevalence (OR = 0.76, p = 0.041), with DMT2 patients showing 24% lower odds of developing RA. Uric acid levels exhibited a significant relationship with RA occurrence (OR = 1.67, p = 0.003), with normal uric acid levels associated with 67% higher odds of RA. Breast cancer status showed a statistically significant association with RA risk (OR = 0.54, p = 0.037), with breast cancer patients demonstrating 46% lower odds of developing RA.”
In Discussion we added: “Similar results obtained in Table 2, when we compared RA with other rheumatic and musculoskeletal diseases, RMD, (1870 patients), we found statistically significant associations between RA and all parameters investigated in this study, except for age and BMD. This finding could be explained by the fact that most of the RMD occur at young and mid-dle-aged females and their RMD put them at increased risk of low BMD. Further analysis within the RA cohort revealed that RA patients had a high prevalence of low BMD (72.3%) and low vitamin D (63%) but high serum uric acid (20.85%). To our knowledge, this is the first study investigating the prevalence of low BMD in Bahraini patients with RA. Our result of low BMD in RA is consistent with a recent study from three northwest European countries that demonstrated low BMD in RA but also an association between low BMD and radiographic RA damage [30]. The coexistence of gout and RA is relatively rare, which could be due to a lack of studies in this area. However, in Bahrain RA and gout have not been reported before, except in one previous study done by our team, which showed that RA patients had hypovitaminosis and high serum uric acid levels and the correction of hypovitaminosis with vitamin D3 therapy led to a reduction of serum uric acid in Bahraini RA patients, nevertheless, that was a small cohort of only 30 RA patients and BMD data was not included in that study [31].”
3.Table 1 is confusing - for example there are 281 RA patients - 268 of whom were female --but how did this account for only 4.8%?
Authors’ reply: Thank you for your insightful feedback regarding Table 1. We apologize for any confusion this may have caused. The percentages in Table 1 represent the proportion of male and female patients within the entirety of the study population, which includes both rheumatoid arthritis (RA) and non-RA patients. We explained that we provide percentage per row not column in our tables in Tables 1, 2, 3. We explained that we provide percentage per column in Table 4.
To clarify, in the RA cohort specifically (as detailed in Table 4), we observe that out of 282 RA patients, 268 are female, accounting for 95.38% of the RA group, while males make up 4.62% (p < 0.001). This distinction highlights the significant female predominance in the RA population compared to the broader cohort presented in Table 1.
